

# A quantitative assessment of the behavior of metallic elements in urban soils exposed to industrial dusts near Dunkirk (Northern France)

Marine Casetta[1], Sylvie Philippe[1], Lucie Courcot[1], David Dumoulin[2], Gabriel Billon[2], François Baudin[3], Françoise Henry[1], Michaël Hermoso[1], Jacinthe Caillaud[1]

[1] LOG, Laboratoire d'Océanologie et de Géosciences, Université du Littoral Côte d'Opale, Université de Lille, CNRS, IRD, UMR 8187, Wimereux, France

[2] Univ. Lille, CNRS, UMR 8516 – LASIRE, Laboratoire Avancé de Spectroscopie pour les Interactions, la Réactivité et l'Environnement, F-59000 Lille, France

[3] Sorbonne Université – CNRS, UMR 7193 ISTeP – Institut des Sciences de la Terre de Paris, 75005 Paris, France

Correspondence to: Marine Casetta (marine.casetta@univ-littoral.fr)

## Abstract

In urban and industrialized areas, soil contamination and degradation caused by the deposition of industrial dusts may pose significant health and environmental risks. This problem relates to the vertical mobility and bioavailability of Potentially Toxic Elements (PTE). This study investigates the fate of PTE brought by industrial dusts in urban soils located in the Dunkerque agglomeration, one of the most industrialized areas of France. Four soil short cores were collected in the city of Gravelines (Dunkerque agglomeration) following a gradient from the industrial emitters to the deposition site. The soil cores were cut into discrete 1-cm-sections to study their PTE concentrations (using ICP-AES/MS analyses). Single HCl extraction was performed to evaluate the mobility of PTE in soils and to discuss their specific behavior according to the current soil parameters. For this purpose, the main soil parameters were identified (grain-size distribution, mineralogy, pH, CEC, TOC, calcium carbonates and water contents) in addition to the soil chemical composition (XRF, ICP-AES/MS analyses). The studied soils revealed globally low absorbent capacities for pollutants (CEC averaging 5.3 meq/100g), partially counterbalanced by the buffering effect of calcium carbonates (contents ranging from 8 % to 30 %). We highlighted minor (1<EF<3) to moderately severe (5<EF<10) Enrichment Factors in industrial PTE (Cr, Ni, Mo, Mn, Cd, Zn), in the first 3 centimeters of the soils located near the industrial emitters. The contamination profiles of these soils are concordant with atmospheric inputs of metallurgical dust. Using a relatively strong leaching reagent (HCl 1M), we estimated a low vertical mobility for Cr, Ni and Mo (average leached ratios <25 %) in soils, suggesting that these industrial PTE mainly occur in refractory phases (natural or anthropogenic). Mn, Cd and Zn, related to industrial and/or urban sources, present a higher mobility (average leached ratios > 60% for Mn and Cd, and averaging 44 % for Zn). Our study points out the stability of industrial PTE in soils under current physicochemical conditions (calcareous soils with a slightly basic pH of 7.8). In this context, the monitoring of industrial PTE in these urban soils is highly



recommended, considering (1) the presence of allotment gardens in the vicinity of the emitters and (2) the potential evolution of soil conditions as a result of increased flooding events.

## 1 Introduction

Soils are essential to the survival of civilizations and terrestrial ecosystems, although they are extremely vulnerable and
nonrenewable at the scale of human life (Minami, 2009; Smith et al., 2015; Stavi et al., 2016). The Food and Agriculture Organization of the United Nations recently recorded 5 670 million hectares of land worldwide in declining biophysical conditions (FAO and UNEP, 2021). This report indicates than human activities are responsible for more than 29 % of this deterioration through change of land uses, soil over-exploitation, climate change, introduction of invasive exotic species or release of pollutants (Emadodin and Bork, 2012; Tetteh, 2015; Dror et al., 2022). Among the various soil pollutants, Potentially
Toxic Elements (PTE) are particularly studied regarding the threat that they represent for the ecosystems and human health when mobilized and transferred to groundwater or to the food chain (Zhuang et al., 2009; Rajmohan et al., 2014; Sun et al., 2018). Soils are at a key interface of several environmental compartments, and their physicochemical properties make them both vectors and reservoirs for contaminants (Girard, 2005; Kandpal et al., 2005; Palansooriya et al., 2020; Sarkar et al., 2021). In urban areas, anthropogenic activities significantly contribute to soil pollution by PTE, especially metallurgical activities and
mostly via atmospheric deposits (Duzgoren-Aydin et al., 2006; Khademi et al., 2019; Manta et al., 2002). The resulting soil degradation by PTE inputs may cause major health and environmental problems related to the increasing density of population (Schulin et al., 2007; Douay et al., 2013; Ortega Montoya et al., 2021).

In the Dunkerque agglomeration (Northern France), over 150 plants facilities constitute a major risk for the environment: in
2021, the local metallurgical industries emitted more than 2 700 t of total dust in the atmosphere, pinpointing concerns about the surrounding soil contamination (Registre des émissions polluantes, 2023). A previous study, based on the spatial distribution of PTE in surficial soils was carried out in the city of Gravelines (Dunkerque agglomeration) (Casetta et al., 2024). This study provided some insights into soil contamination by industrial dust, highlighting (1) an appreciable portion of coals, iron ores, slags and other metallurgical products (>88 %) in the industrial dust falling in the streets of the city during NE wind
and dry periods, (2) a diffuse contamination of soils by PTE associated to industrial dust, such as Cr, Ni, Mo particularly but also Zn, Cd, Mn, and (3) the punctual degradation of the soil quality by industrial dust. Contamination metals are generally considered as more mobile in soils than lithogenic ones (Wieczorek et al., 2023). Considering that sources of contamination are multiple, but that some PTE are characteristics of local and industrial dust, the main question concerns their vertical mobility. Do the input of industrial dusts in soils represent a potential impact for solution, and then for biota? A further question
addressed relates to the soil capacity to retain metals potentially released from contamination particles.



The present study aims at evaluating the behavior of PTE associated with industrial dust or other anthropogenic inputs in selected urban soils (Gravelines), with depth and at a centimetric scale. For this purpose, HCl single extractions of soil samples were performed. The physicochemical profiles (grain size distribution, water contents, pH, CEC, calcium carbonates, total

organic carbon, major oxide concentrations, mineralogical composition) of four soil short cores (0-11 cm) were established for better constrain the interpretation of the behavior of PTE with respect to the soil matrix. The vertical distribution of some PTE (Cr, Mn, Ni, Cu, Zn, Mo, Cd) in soils was determined before and after simulated leaching to highlight specific behavior of the considered PTE and to discuss a potential link with the industrial provenance.

## 2 Material & methods

### 2.1 Study area and sampling

The city of Gravelines (50°59' N, 2°08' E) is located in Northern France on the North Sea coast (figure 1). The climate is temperate: average annual rainfall of 727 mm and annual temperature ranging between 2°C and 22°C (MERRA-2 meteorological data, 1980-2016). The city has a surface area of 22.66 km$^2$ and its population was 11 014 in 2019 (INSEE). The land uses distribution is 52 % of urbanized/industrialized spaces, 22 % of agricultural, 16 % of natural spaces and 10 %

of infrastructures. Three metallurgical production sites and one plant for receiving, handling and storing ores in open air were installed less than five km from the city center as result of the construction of the seaport of Dunkerque. Marine and alluvial clay and sandy sediments were deposited during the Holocene on the study area, belonging to the Flanders coastal plain (Leplat et al., 1988). The soils mainly consist of clayey-sandy matrials and are rich in calcium carbonates (median value: 13 %) (Sterckeman et al., 2004). According to the pedological classification of the French Association for Soils Study, the soils of

Gravelines can be ascribed to "Thalassosols", characteristic of a pedogenetic evolution on marine formations (Baize and Girard, 2009; GIS Sol and RMT Sols et Territoires, 2019). In the highly populated studied area, the coastal plain is drained through a dense network of channels to avoid flooding from the sea, or by brackich/freshwater. Thus, the characteristics of the studied soils may locally significantly differ from the general type proposed.

Four sampling stations were selected in the city of Gravelines (Fig. 1) according to: (1) the concentrations in PTE associated to industrial dust fallout (Casetta et al., 2024); (2) the nature of the soil matrices; and (3) their distance to the emission sources. These selected stations present 2 types of uses: collective use as parks and green spaces in the city center (station 1), and mesophilic grassland or deciduous planting trees located near the industrial emitters (stations 2, 3 and 4). Short cores were collected at the four stations in July 2021 during the summer season using a manual auger and PVC tubes. For each site, one

soil core (diameter of 4.5 cm and depth of 11 cm, in order to consider study the upper part of the A horizon) was taken (herbaceous formations) and cut into 1 cm sections using a Teflon core cutting table. Thus, a total of 44 soil samples were analyzed.





**Figure 1: Map of the city of Gravelines indicating the sampling stations located southwest of the industrialized seaport (IGN – BD TOPO® Nord 2023).**

## 2.2 Nature of the soil matrices

### 2.2.1 General soil parameters

The water content of the soils was determined by weighing samples, before and after drying a 30-g-aliquot at 65°C in a ventilated oven (according to the normative protocol NF EN 1097-5). A combined glass electrode and a pH-Meter were used to measure soil pH in deionized water (1:5 soil: solution) (NF ISO 10390, see Goix et al., 2015). The soil cation exchange



capacity (CEC) was determined using a spectrophotometric method based on cobaltihexamine chloride absorbance (Aran et al., 2008). Aliquots of 2 g of dry samples were mixed to a 0.01 N solution of cobaltihexamine chloride during 1 hour. The solution was filtered through 0.22 µm filters (cellulose acetate) after 10 minutes centrifugation (4000 g). The CEC was obtained

by measuring the absorbance at 472 nm of the collected mixture.

The grain-size distribution and calcium carbonate contents were measured on the subsurface (0-1 cm), the middle (4-5 cm) and in a deeper layer (9-10 cm) for the four sampled cores. The grain-size distribution was determined on wet samples by laser diffraction. The Coulter LS 13 320 instrument (gallium arsenide, 750 nm wavelength, Brea, USA) measured a particle diameter ranging from 0.375 to 2000 µm. Prior to analysis, organic matter was removed by adding 50 mL of $H_2O_2$ (35 %), according to

the Belgian standard NBN 589-207 §3 (Leifeld and Kögel-Knabner, 2001; Amar et al., 2021). A 2 mm mesh sieve was used to remove the higher size fraction (roots, stems and leaves) and each soil sample was sonicated to achieved disaggregation. Particle size classes were assigned according to the Soil Science Division Staff (2017) grain-size scale, i.e. 0.375–2.0 µm for the clay fraction, 2.0–20 µm for the fine silt fraction, 20–50 µm for the coarse silt fraction and 50–2000 µm for the sand

fraction. A manual calcimeter (OFITTE, Houston, USA; dried and grinded samples) was calibrated with pure calcium carbonate to obtain the total carbonate content ($CaCO_3$) of the soil samples. This content was estimated by the gas pressure emitted after reaction of 1 g of fine powdered sample when adding 20 mL of HCl 20 %. Measurement uncertainties were estimated from three analytical replicates performed on the highest and lowest values during laboratory phases. The absolute uncertainties for the studied soil parameters are as follows: pH ± 0.1; CEC ± 0.2 meq/100 g; $CaCO_3$ content ± 1 %; particle

size distribution: clay ± 0.5 %; fine silt ± 2 %; coarse silt ± 2 %; sand ± 3 %.

**2.2.2 Total organic carbon (TOC) content and characterization by Rock-Eval measurements**

The soil samples were analyzed with a Rock-Eval 6 (RE6) Turbo device (Vinci Technologies, Nanterre, France) using the basic setup for soil organic matter analysis (Disnar et al., 2003; Hetényi and Nyilas, 2014). The RE6 technique required two steps: (1) the pyrolysis of 60 mg of finely ground soil (<250 µm) in an $N_2$ atmosphere (from 200°C to 650°C; heating rate of

30°C min⁻¹); followed by (2) the oxidation of the pyrolysis residues in an oxygenated atmosphere (from 300°C to 850°C; heating rate of 20°C min⁻¹). Volatile hydrocarbon (HC) effluents during pyrolysis were detected and quantified using Flame Ionization Detection (FID), while oxygen compounds (CO, $CO_2$) were quantified during the two steps by infrared detection. Measurements resulted in the production of five thermograms (S1 to S5) per sample, corresponding to free hydrocarbons (S1), pyrolyzable hydrocarbons (S2), $CO_2$ and CO (S3) generated during pyrolysis step, and to CO and $CO_2$ (S4 for organic residual

carbon and S5 for mineral carbon) produced during the oxidation step. The complete description of the method is available in Lafargue et al. (1998) and Cécillon et al. (2018). Finally, the analysis of the different thermograms allowed the calculation of several parameters (Espitalié et al., 1977; Vandenbroucke and Largeau, 2007; Espitalié et al., 1985), as:

-    The Total Organic Carbon content (TOC; wt %) corresponding to the sum of residual and pyrolyzed organic carbon;
-    The Hydrogen Index (HI; mg HC/g TOC) corresponding to the quantity of HC released relative to TOC (S2/TOC);



- The Oxygen Index (OI$_{RE6}$; mg O$_2$/g TOC) corresponding to the quantity of oxygen released as CO and CO$_2$ during pyrolysis and relative to TOC and calculated as follows (Lafargue et al., 1998):

$$OI_{RE6} = \left[ \left( \frac{16}{28} \times OI_{CO} \right) + \left( \frac{32}{44} \right) \times OI_{CO2} \right]$$

with OI$_{CO}$ = 100 x S3$_{CO}$/TOC and OI$_{CO2}$ = 100 x S3$_{CO2}$/TOC

HI and OI$_{RE6}$ are used to highlight the main type of organic matter present in the studied soils. For this purpose, these index are compared to the Van Krevelen diagram (H/C vs. O/C) of Espitalié et al., 1977.

### 2.2.3 Major oxide measurement

An energy dispersive X-ray fluorescence spectrometer (Bruker S2 PUMA ED-XRF) was used to measure the concentration of 7 major oxides (Na$_2$O, MgO, Al$_2$O$_3$, SiO$_2$, K$_2$O, CaO, Fe$_2$O$_3$) on fused soil beads. The spectrometer was equipped with a 50

kV Ag anode tube (maximum power: 50 W; maximum high voltage: 50 kV; maximum current: 2 mA; cooling medium: air) and a high-resolution Silicon Drift Detector (<141 eV for Mn-Kα1). Spectral data were analyzed by Spectra Elements software version 2.0. Soil samples were dried, sieved (2 mm), ground, homogenized and calcined at 1050°C (loss on ignition: LOI). Then, fused bead specimens were formed by mixing 500 to 1 000 mg of calcinated powder with a lithium tetraborate (33 %), lithium metaborate (67 %) and lithium bromide (<1 %) mixture. These flux mixtures were loaded in a Katanax® K1 Prime

fluxer and heated up to 1060 °C during 20 minutes. Analytical quality of the XRF measurements was controlled by analyzing 11 certified samples of stream sediments. Determination limits (DL) and measurement uncertainties are available in Table S1 in the supplementary materials.

### 2.2.4 Mineralogy

Mineralogy X-ray diffraction (XRD) studies were carried out using an AXS D4 Endeavor Diffraction System (Bruker; 35 kV,

30 mA; Cu K$_α$ radiations) coupled to a PSD LynxEye detector. Crushed soil samples were examined as random total powder to obtain the total mineralogical composition between 3 and 60°2θ. Background stripping, diffraction peak indexing, mineral identification by comparison with the files of the Joint Committee on Powder Diffraction Standards (JCPDS) and semi-quantitative analysis were carried out using the X'Pert data HighScore software. Thus, a list of four minerals (Table S2 in the supplementary materials) was chosen according to: (1) the best match between the positions of peaks (score); and (2) the

quality of the reference based on a calibration with titanium (Reference Intensity Ratio, RIR). For the clay preparation, 0.2 M hydrochloric acid was used to decalcify the soil samples and the excess acid was removed by repeated centrifugations after rinsing with deionized water. Settling was used to isolate the clay-sized fraction (<2 µm) which was next oriented on glass slides (oriented mounts). Four clay minerals are identified between 2.5 and 32°2θ according to the position of the (001) series of basal reflections on air-dried, glycolated (after saturation for 12h in ethylene glycol) and 490°C-heated (for 2 hours)

diffractograms (Holtzapffel, 1985). Their semi-quantification were carried out on the glycerol curve using MacDiff software.



The reproducibility of technical works and measurements was tested and the relative error was <5 % (Bout-Roumazeilles et al., 1999).

## 2.3 Vertical evolution of PTE concentrations

### 2.3.1 PTE concentrations in total and leached soil samples

An amount of 200 mg of each ground soil samples were digested using first a concentrated HF–HNO$_3$ mixture (67:33 v:v) and then 2 others concentrated HCl–HNO$_3$ mixture (67:33 v:v). Each digestion step lasted 48 hours at 125°C before evaporation. Dissolved samples were finally diluted in 9 mL with acidified ultrapure MilliQ® (Millipore 18.2 MΩ.cm resistivity) water. Each solution was filtered on 0.22 µm filters (cellulose acetate) to remove potential residues. All the reagents were of the optimal/suprapur grade. Single extractions were performed on the same ground soil samples by mixing 1 g of powder with 20

mL of cold 1 M HCl for 24 h (Billon, 2001; Philippe et al., 2008). Next, leached samples were centrifuged and the supernatant was filtered on 0.22 µm filters (cellulose acetate) for ICP analyses. According to Hamdoun et al. (2015) and Yu et al. (2021), this technique allows the estimation of the general mobility and reactivity of PTE in the soil matrices. Furthermore, the use of 1 M hydrochloric acid aims to limit under-estimation of PTE mobility related to the buffer capacity of the carbonated soils of Gravelines (Birch, 2017). The total content of the studied PTE was determined both in total and leaching solutions using ICP-

AES (Agilent 5110 VDV, for Cu) and ICP-MS (Agilent 7850, for Cr, Mn, Ni, Zn, Mo and Cd). The accuracy and precision were controlled using two sediment standard reference materials (MESS-3 and PACS-2), twelve analytical triplicates and six blank samples. The recovery values of the reference standards and the detection limits are available in Table S3 in the supplementary materials.

### 2.3.2 PTE Enrichment Factors (EF)

Trace metal concentrations were compared to those of agricultural ploughed soils of the French Flemish Coastal plain (Wateringues marine plain), supposed to be preserved from potential contamination sources (because distant from industrial activities, busy roads, houses) (Sterckeman et al., 2004). The EF evaluates the degree of metallic contamination of soils by distinguishing anthropogenic from natural metal concentrations (Ye et al., 2011; Harb et al., 2015). Aluminium (Al) is commonly used as a normalizing element (Brady, 1984; Duodu et al., 2017). In this sense, EF were calculated following

Eq. (1):

$$EF = (C_n/C_{Al})/(B_n/B_{Al}) \quad (1)$$

where $C_n$ and $C_{Al}$ are the concentration of a metal element n and the concentration of Al in the sample, respectively; $B_n$ and $B_{Al}$ are the concentration of a metal element n and the concentration of Al in the Wateringues marine plain background (mean

values of the agricultural soils from (Sterckeman et al., 2004)), respectively.



## 2.4 Data analysis

Maps were made using QGIS 3.10 (QGIS Development Team, 2023). Statistical computing and graphics were performed on R software (R Core Team, 2022) using the following packages: FactoMineR (Lê et al., 2008), ggplot 2 (Wickham, 2016), corTest (Yu et al., 2020), factoextra (Kassambara and Mundt, 2020) and corrplot (Wei and Simko, 2021). As suggested by

Chapman (1996), all concentration values below the detection limit (DL) were replaced by half of the DL for the statistical analyses. As the collected data present a non-parametric distribution, the correlations between results were highlighted using the Spearman's correlation test (R).

## 3 Results

### 3.1 Soil cores properties

#### 3.1.1 Physicochemical parameters

The physicochemical properties of the soil cores are available in Table 1. The study of soil parameters reveals significant variations between the four sampled stations and along depth profiles. Soil pH from all sampled stations is slightly basic (range: 7.4 – 8.3; average value: 7.8). Notable heterogeneity in soil textures is highlighted between cores. Core 1 presents a "sandy loam" texture with the highest sand contents (40 % to 58 %) and the lowest clay contents (9 % to 12 %). Conversely, cores 3

and 4 are characterized by a "silt loam" texture, higher clay contents (13 % to 19 %) and lower sand values (8 % to 26 %). Core 2 presents an intermediate "loam" texture with clay and sand values ranging from 12 % to 16 % and from 31 % to 41 %, respectively (Richer-De-Forges et al., 2008). No important vertical variation is observed in the grain-size distribution of soil cores, except in core 4 in which clay content increases in subsurface in an inverse proportion to sand content.

**Table 1: Physicochemical characteristics of the sampled cores. SD: standard deviation and "-" means no data. TOC: Total Organic Carbon; CEC: Cationic Exchange Capacity; LOI: Loss On Ignition at 1050°C. Detection Limits are available in table S1. Station 1: park and green spaces; Stations 2, 3 and 4: mesophilic grassland or deciduous planting trees.**

| Core | Depth (cm) | pH | Water content (%) | TOC (%) | CEC (meq /100 g) | CaCO$_3$ (%) | Clay (%) | Fine silt (%) | Coarse silt (%) | Sand (%) | LOI (%) | Na$_2$O (%) | MgO (%) | Al$_2$O$_3$ (%) | SiO$_2$ (%) | K$_2$O (%) | CaO (%) | Fe$_2$O$_3$ (%) |
|---|---|---|---|---|---|---|---|---|---|---|---|---|---|---|---|---|---|---|
| | 0-1 | - | 28 | 2.6 | 4.6 | 8 | 11 | 29 | 7 | 54 | 9 | 0.5 | <0.7 | <1.9 | 83 | 1.2 | 4.0 | 1.1 |
| | 1-2 | - | 31 | 3.1 | 5.1 | - | - | - | - | - | 11 | <0.5 | <0.7 | <1.9 | 81 | 1.2 | 4.0 | 1.1 |
| | 2-3 | 7.8 | 32 | 3.4 | 5.4 | - | - | - | - | - | 11 | 0.6 | <0.7 | <1.9 | 80 | 1.1 | 4.5 | 1.1 |
| 1 | 3-4 | - | 32 | 3.5 | 5.5 | - | - | - | - | - | 11 | 0.5 | <0.7 | <1.9 | 81 | 1.1 | 4.2 | 1.1 |
| | 4-5 | 7.7 | 31 | 3.1 | 5.8 | 8 | 12 | 37 | 12 | 40 | 10 | <0.5 | <0.7 | <1.9 | 82 | 1.2 | 4.0 | 1.1 |
| | 5-6 | - | 29 | 2.9 | 5.6 | - | - | - | - | - | 10 | 0.7 | <0.7 | <1.9 | 81 | 1.1 | 4.0 | 1.1 |
| | 7-8 | - | 25 | 2.5 | 5.2 | - | - | - | - | - | 9 | 0.6 | <0.7 | <1.9 | 83 | 1.2 | 4.2 | 1.1 |





| Core | Depth (cm) | pH | Water content (%) | TOC (%) | CEC (meq /100 g) | CaCO$_3$ (%) | Clay (%) | Fine silt (%) | Coarse silt (%) | Sand (%) | LOI (%) | Na$_2$O (%) | MgO (%) | Al$_2$O$_3$ (%) | SiO$_2$ (%) | K$_2$O (%) | CaO (%) | Fe$_2$O$_3$ (%) |
|---|---|---|---|---|---|---|---|---|---|---|---|---|---|---|---|---|---|---|
| | 9-10 | 7.7 | 24 | 2.3 | 5.7 | 9 | 9 | 27 | 6 | 58 | 9 | <0.5 | <0.7 | <1.9 | 82 | 1.3 | 4.5 | 1.2 |
| | 10-11 | - | 23 | - | 5.0 | - | - | - | - | - | 9 | 0.7 | <0.7 | <1.9 | 82 | 1.4 | 4.6 | 1.1 |
| | Average | 7.7 | 28 | 2.9 | 5.3 | 8 | 11 | 31 | 8 | 51 | 10 | 0.6 | <0.7 | <1.9 | 82 | 1.2 | 4.2 | 1.1 |
| | SD | 0.04 | 4 | 0.4 | 0.4 | 1 | 1 | 5 | 3 | 10 | 1 | 0.1 | - | - | 1 | 0.1 | 0.3 | 0.0 |
| | 0-1 | 7.4 | 29 | 2.5 | 4.8 | 11 | 16 | 41 | 12 | 31 | 11 | 0.6 | <0.7 | 2.2 | 76 | 1.3 | 6.4 | 1.9 |
| | 1-2 | - | 29 | 2.4 | 5.0 | - | - | - | - | - | 11 | <0.5 | <0.7 | 2.6 | 77 | 1.3 | 6.4 | 1.8 |
| | 2-3 | - | 28 | 2.2 | 4.8 | - | - | - | - | - | 10 | <0.5 | <0.7 | 2.6 | 76 | 1.5 | 6.7 | 1.9 |
| | 3-4 | - | 22 | 1.6 | 4.4 | - | - | - | - | - | 10 | <0.5 | <0.7 | 2.9 | 77 | 1.3 | 6.7 | 1.9 |
| 2 | 4-5 | 7.8 | 20 | 1.3 | 3.8 | 12 | 17 | 34 | 12 | 38 | 9 | <0.5 | <0.7 | 2.8 | 78 | 1.2 | 6.7 | 1.9 |
| | 5-6 | - | 18 | 1.1 | 3.8 | - | - | - | - | - | 9 | <0.5 | <0.7 | 2.3 | 78 | 1.5 | 6.8 | 2.0 |
| | 7-8 | - | 18 | 1.0 | 3.6 | - | - | - | - | - | 9 | <0.5 | <0.7 | 2.9 | 74 | 1.2 | 6.7 | 1.9 |
| | 9-10 | 8.0 | 17 | 1.0 | 3.7 | 12 | 12 | 33 | 14 | 41 | 8 | - | - | - | - | - | - | - |
| | 10-11 | - | 16 | - | 3.5 | - | - | - | - | - | 8 | 0.5 | <0.7 | 2.4 | 79 | 1.3 | 6.9 | 2.0 |
| | Average | 7.7 | 22 | 1.7 | 4.2 | 12 | 15 | 36 | 13 | 36 | 9 | 0.6 | <0.7 | 2.6 | 77 | 1.3 | 6.7 | 1.9 |
| | SD | 0.3 | 5 | 0.7 | 0.6 | 1 | 2 | 5 | 2 | 5 | 1 | 0.1 | - | 0.3 | 2 | 0.1 | 0.2 | 0.0 |
| | 0-1 | - | 57 | 4.7 | 7.3 | 25 | 14 | 41 | 20 | 26 | 23 | <0.5 | <0.7 | 4.2 | 53 | 1.4 | 14.8 | 2.6 |
| | 1-2 | 7.5 | 49 | 4.1 | 7.2 | - | - | - | - | - | 21 | <0.5 | <0.7 | 4.5 | 57 | 1.4 | 16.8 | 2.8 |
| | 2-3 | - | 45 | 3.2 | 6.7 | - | - | - | - | - | 20 | 0.6 | <0.7 | 4.7 | 54 | 1.5 | 15.7 | 2.6 |
| | 3-4 | - | 40 | 2.7 | 6.2 | - | - | - | - | - | 19 | 0.6 | <0.7 | 4.1 | 55 | 1.4 | 16.1 | 2.7 |
| 3 | 4-5 | 8.0 | 35 | 2.1 | 5.8 | 28 | 15 | 38 | 22 | 25 | 19 | <0.5 | <0.7 | 4.5 | 56 | 1.4 | 16.4 | 2.8 |
| | 5-6 | - | 31 | 1.7 | 5.4 | - | - | - | - | - | 18 | 0.5 | <0.7 | 4.2 | 54 | 1.5 | 16.4 | 2.7 |
| | 7-8 | - | 28 | 1.2 | 5.0 | - | - | - | - | - | 17 | 0.8 | <0.7 | 4.8 | 56 | 1.5 | 16.6 | 2.8 |
| | 9-10 | 8.0 | 26 | 1.1 | 4.8 | 30 | 16 | 42 | 18 | 24 | 17 | 0.5 | <0.7 | 5.6 | 56 | 1.4 | 16.8 | 2.7 |
| | 10-11 | - | 26 | - | 4.7 | - | - | - | - | - | 16 | <0.5 | <0.7 | 4.4 | 54 | 1.5 | 15.2 | 2.7 |
| | Average | 7.8 | 37 | 2.6 | 5.9 | 28 | 15 | 41 | 20 | 25 | 19 | 0.6 | <0.7 | 4.5 | 55 | 1.4 | 16.1 | 2.7 |
| | SD | 0.3 | 11 | 1.3 | 1.0 | 3 | 1 | 2 | 2 | 1 | 2 | 0.1 | - | 0.5 | 1 | 0.1 | 0.7 | 0.1 |
| | 0-1 | - | 46 | 3.3 | 6.8 | 15 | 19 | 54 | 19 | 8 | 15 | <0.5 | <0.7 | 4.2 | 66 | 1.4 | 9.6 | 2.6 |
| | 1-2 | 7.8 | 43 | 3.1 | 6.7 | - | - | - | - | - | 15 | <0.5 | <0.7 | 4.0 | 67 | 1.4 | 9.5 | 2.6 |
| | 2-3 | - | 40 | 2.7 | 6.5 | - | - | - | - | - | 14 | 0.6 | <0.7 | 4.1 | 66 | 1.3 | 9.6 | 2.6 |
| | 3-4 | - | 36 | 2.2 | 6.2 | - | - | - | - | - | 13 | 0.6 | <0.7 | 4.1 | 67 | 1.4 | 9.6 | 2.7 |
| 4 | 4-5 | 7.8 | 31 | 1.6 | 5.7 | 15 | 16 | 48 | 23 | 13 | 13 | 0.5 | <0.7 | 4.2 | 68 | 1.6 | 10.2 | 2.7 |
| | 5-6 | - | 23 | 1.3 | 5.6 | - | - | - | - | - | 12 | 0.7 | <0.7 | 2.9 | 75 | 1.3 | 6.4 | 1.6 |
| | 7-8 | - | 22 | 1.1 | 5.3 | - | - | - | - | - | 12 | <0.5 | <0.7 | 4.7 | 69 | 1.4 | 10.2 | 2.7 |
| | 9-10 | 8.3 | 19 | 1.0 | 5.2 | 15 | 13 | 41 | 24 | 24 | 12 | 0.6 | <0.7 | 4.5 | 68 | 1.6 | 10.3 | 2.7 |
| | 10-11 | - | 19 | - | 5.2 | - | - | - | - | - | 11 | 0.5 | <0.7 | 4.7 | 68 | 1.6 | 10.1 | 2.8 |
| | Average | 8.0 | 31 | 2.0 | 5.9 | 15 | 16 | 48 | 22 | 15 | 13 | 0.6 | <0.7 | 4.2 | 68 | 1.5 | 9.5 | 2.6 |
| | SD | 0.30 | 11 | 0.9 | 0.7 | 0 | 3 | 7 | 3 | 8 | 2 | 0.1 | - | 0.5 | 3 | 0.1 | 1.2 | 0.3 |
| | Total average | 7.8 | 30 | 2.3 | 5.3 | 16 | 14 | 39 | 15 | 32 | 13 | 0.6 | - | 3.8 | 70 | 1.4 | 9.2 | 2.1 |
| | SD | 0.2 | 10 | 1.0 | 1.0 | 8 | 3 | 8 | 6 | 15 | 4 | 0.1 | - | 0.9 | 11 | 0.1 | 4.6 | 0.7 |

TOC content shows little variation between cores (average value: 2 %; standard deviation: 1 %) and globally decreases with depth. The highest value is measured in the subsurface of core 3 (4.7 %). Core 1 presents another pattern and remains relatively

stable along the profile (from 2.3 to 3.5 %). The OI and HI records (Table S4 in the supplementary materials) present slight differences between the subsurface and the deepest layer of soil cores 2, 3 and 4 (HI values ranging from 152 to 335 mg HC/g





TOC; $OI_{RE6}$ values ranging from 154 to 218 mg $O_2$/g TOC). Cores 3 and 4 follow a relatively close trend with intermediate values of HI and higher values of $OI_{RE6}$. Core 1 exhibits a similar pattern to core 2 for $OI_{RE6}$ but has the highest HI values.

The CEC values range from 3.5 to 7.3 meq/100 g and are globally low, according to Rengasamy and Churchman (1999). Maxima CEC values are measured in the subsurface of cores 3 and 4 (7.3 and 7 meq/100 g, respectively) while minimum CEC is measured in core 2 [10-11cm] (3.5 meq/100 g). $CaCO_3$ contents present no significant variation with depth but are higher in core 3 (average: 28 %) and 4 (average: 15 %). The lowest $CaCO_3$ values are observed in core 1 (average: 8 %). The average water content by core can be sorted as follows: core 3 (37 %) > core 4 (31 %)> core 1 (28 %) > core 2 (22 %). While it presents homogeneous vertical profiles in cores 1 and 2, the water content significantly decreases with depth in cores 3 (57 % to 26 %)

and 4 (46 % to 19 %). Major oxide measurements are presented in Table 1. $SiO_2$, CaO and $Fe_2O_3$ contents appear stable with depth but present significant variations between cores. The highest $SiO_2$ contents are observed in cores 1 and 2 (>78 %). The lowest are notable in core 3 (<57 %) where $Al_2O_3$ and CaO concentrations follow an inverse pattern ($Al_2O_3$ > 4.2 % and CaO > 15.2 %). $Fe_2O_3$ contents globally range from 1.9 % and 3.0 % except on core 1 (average: 1.1 %).

### 3.1.2 Total and clayey mineralogical composition

Figure 2 presents the vertical profile of total and clayey mineralogical composition in the four soil cores (the complete data set is available in Table S5 in the supplementary materials). The study of total minerals indicates a dominance of quartz (>50 %) in all cores and no significant evolution of their composition downcore. Core 1 is characterized the highest quartz values (>70 %) and the lowest feldspars, calcite and micas concentrations, with average concentrations of 3 %, 4 % and 9 %, respectively. Core 3 has a different behavior with the highest calcite and micas concentrations (22 % and 20 %) and lowest

quartz and felspars values (54 % and 5 %). A predominance of quartz is also observed on cores 2 and 4 (range: 67 % to 81 %). The same proportions of calcite, micas and feldspars are measured in these two cores (averaging 10 %, 12 % and 4 %, respectively). The clay minerals nature differed from core 1 to core 4 with an increasing percentage of smectite and a decreasing percentage of all the others (illite, kaolinite, chlorite). While illite dominates the assemblage of the clay minerals in cores 1 and 2 (> 41 %), smectite is the preponderant clay mineral in cores 3 and 4 (> 50 %). All the soil cores present higher smectite

percentages with increasing depth (range: 34 % to 67 %) and higher illite (32 % to 56 %), kaolinite (14 % to 22 %) and chlorite (9 % to 15 %) percentages in the subsurface (<3 cm).



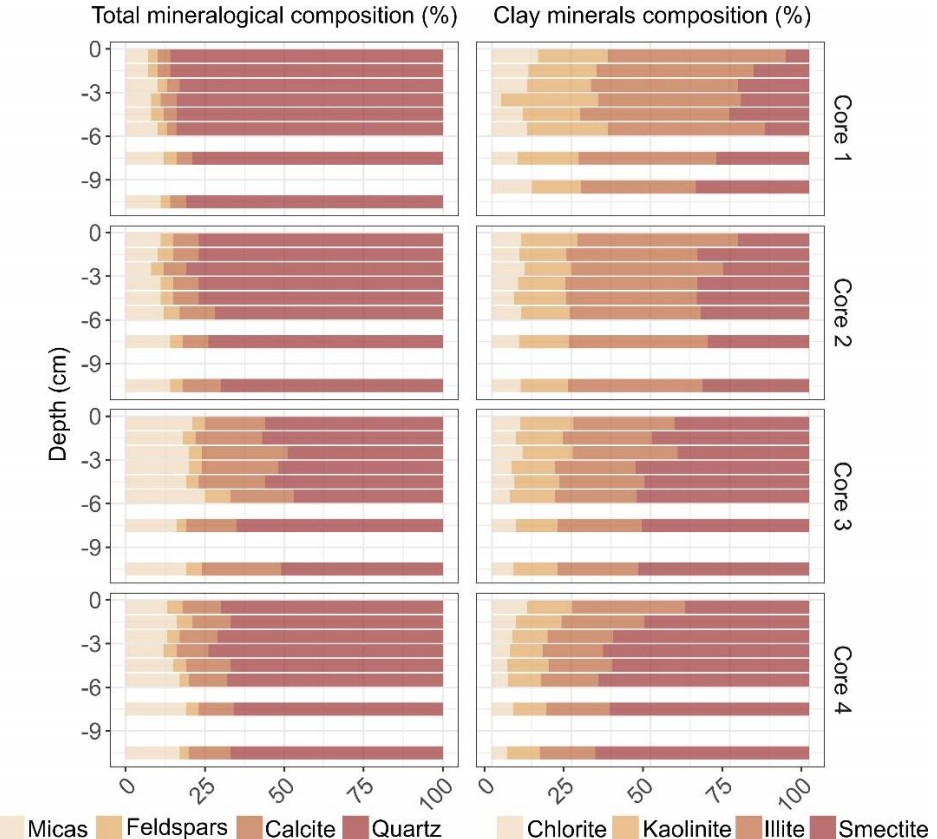

**Figure 2: Vertical profiles of total (left) and clay (right) minerals on the four studied cores.**

The physicochemical parameters of soils (Table 1) support the pedological classification of the upper soils of Gravelines as "Thalassossols", as a global loamy texture was identified in the samples (relatively close to the ploughed soils of the Wateringues marine plain (Sterckeman et al., 2004)). In addition, $CaCO_3$ and CaO concentrations are highly correlated (R= 0.97), suggesting a dominance of calcium as carbonates in soils. This hypothesis is supported by both the pH values that reflect neutral to slightly basic soils (Luo et al., 2015) and the global mineralogical composition of the studied soils.

**3.2 Vertical PTE concentrations in soil cores**

**3.2.1 Total PTE concentrations**

PTE concentrations, measured along the four soil cores, are presented in Table 2. Results indicate variations in PTE concentrations between the studied cores, and higher concentrations in some PTE compared to the ploughed soils of the so-called Wateringues marine plain (Sterckeman et al., 2004). Core 1 presents the lowest average concentrations for all the

analyzed PTE, except Mo (0.4 mg.kg⁻¹). A significant vertical evolution is only observed for this element, increasing from 0.3 mg.kg⁻¹ in the deepest layer to 0.5 mg.kg⁻¹ in the subsurface. For the other cores, all the PTE concentrations are higher in the



subsurface. Core 2 exhibits intermediate average values of 34, 257 and 10 mg.kg[-1] for Cr, Mn and Ni, respectively. This core is also characterized by the highest average Cu concentrations (10 mg.kg[-1]) and the lowest average Mo concentrations (0.3 mg.kg[-1]). Core 3 is characterized by the highest average concentrations in Cr, Mn, Ni, Mo and Zn (with 57, 326, 17, 0.8 and

60 mg.kg[-1], respectively) and the lowest average concentrations in Cu (7 mg.kg[-1]). Cr, Ni, Mo and Zn values are more than twice higher than the Wateringues marine plain background (Sterckeman et al., 2004) in the subsurface of this core. Finally, core 4 presents the same pattern as core 3 in average, but with slightly lower values. Values more than twice higher than the Wateringues marine plain background were observed only for Mo and Zn with concentrations higher than 0.3 and 71 mg.kg[-1], respectively.


**Table 2: Evolution of PTE contents with depths of the four soil cores collected in Gravelines (mg.kg[-1]). Values more than twice higher than the Wateringues marine plain background are denoted in bold (median values). SD: Standard Deviation. N.d.: not determined; "-" means no data. (a) Sterckeman et al., 2004.**

| Core | Depth (cm) | Cr | Mn | Ni | Cu | Zn | Mo | Cd |
|------|-----------|-----|-----|-----|-----|-----|-----|------|
| 1 | 0-1 | 22 | 102 | 8 | 7 | 39 | **0.5** | 0.16 |
| | 1-2 | 22 | 103 | 8 | 8 | 41 | **0.5** | 0.16 |
| | 2-3 | 24 | 109 | 8 | 8 | 45 | **0.5** | 0.17 |
| | 3-4 | 22 | 107 | 8 | 8 | **72** | **0.4** | 0.19 |
| | 4-5 | 22 | 103 | 8 | 8 | 44 | **0.4** | 0.17 |
| | 5-6 | 21 | 105 | 8 | 8 | 41 | **0.4** | 0.18 |
| | 7-8 | 21 | 105 | 8 | 8 | 41 | 0.3 | 0.20 |
| | 9-10 | - | - | - | - | - | - | - |
| | 10-11 | 24 | 106 | 8 | 8 | 40 | 0.3 | 0.18 |
| | Average | 22 | 105 | 8 | 8 | 45 | 0.4 | 0.18 |
| | SD | 1 | 2 | 0.2 | 0.3 | 10 | 0.1 | 0.01 |
| 2 | 0-1 | 37 | 278 | 11 | 11 | 67 | **0.6** | 0.39 |
| | 1-2 | 36 | 265 | 11 | 10 | 63 | **0.5** | 0.34 |
| | 2-3 | 30 | 276 | 10 | 11 | 63 | 0.2 | 0.33 |
| | 3-4 | 33 | 250 | 10 | 12 | 64 | 0.2 | 0.33 |
| | 4-5 | 34 | 254 | 10 | 11 | 56 | 0.2 | 0.31 |
| | 5-6 | 35 | 255 | 10 | 9 | 54 | 0.3 | 0.34 |
| | 7-8 | 33 | 240 | 10 | 9 | 55 | 0.2 | 0.32 |
| | 9-10 | - | - | - | - | - | - | - |
| | 10-11 | 33 | 240 | 9 | 9 | 47 | 0.2 | 0.33 |
| | Average | 34 | 257 | 10 | 10 | 59 | 0.3 | 0.34 |
| | SD | 2 | 14 | 0.5 | 1 | 6 | 0.1 | 0.02 |
| 3 | 0-1 | **67** | 381 | **22** | 9 | **90** | **1.4** | 0.42 |
| | 1-2 | **63** | 335 | **19** | 8 | **73** | **0.9** | 0.42 |
| | 2-3 | **60** | 333 | **18** | 8 | 68 | **0.9** | 0.40 |
| | 3-4 | **57** | 322 | 17 | 8 | 58 | **0.7** | 0.39 |
| | 4-5 | **57** | 321 | 16 | 7 | 53 | **0.7** | 0.38 |
| | 5-6 | 55 | 308 | 16 | 7 | 50 | **0.6** | 0.33 |
| | 7-8 | 52 | 299 | 15 | 7 | 47 | **0.6** | 0.34 |
| | 9-10 | - | - | - | - | - | - | - |
| | 10-11 | 49 | 308 | 14 | 6 | 45 | **0.4** | 0.30 |
| | Average | 57 | 326 | 17 | 7 | 60 | 0.8 | 0.37 |
| | SD | 5 | 24 | 2 | 1 | 14 | 0.3 | 0.04 |





| Core | Depth (cm) | Cr | Mn | Ni | Cu | Zn | Mo | Cd |
|---|---|---|---|---|---|---|---|---|
| 4 | 0-1 | 42 | 300 | 13 | 9 | 56 | **0.6** | 0.24 |
| | 1-2 | 41 | 281 | 13 | 9 | **77** | **0.6** | 0.23 |
| | 2-3 | 41 | 282 | 12 | 9 | 47 | **0.5** | 0.24 |
| | 3-4 | 41 | 292 | 12 | 9 | 45 | **0.4** | 0.23 |
| | 4-5 | 45 | 292 | 12 | 9 | 43 | **0.4** | 0.23 |
| | 5-6 | 40 | 290 | 12 | 9 | 43 | 0.3 | 0.21 |
| | 7-8 | 38 | 279 | 11 | 8 | 40 | 0.3 | 0.21 |
| | 9-10 | - | - | - | - | - | - | - |
| | 10-11 | 38 | 274 | 12 | 8 | 40 | 0.2 | 0.20 |
| | Average | 41 | 286 | 12 | 9 | 49 | 0.4 | 0.22 |
| | SD | 2 | 8 | 0.5 | 0.5 | 11 | 0.1 | 0.01 |
| | Total average | 39 | 244 | 12 | 9 | 53 | 0.5 | 0.28 |
| | SD | 13 | 86 | 4 | 1 | 13 | 0.3 | 0.09 |
| | Wateringues marine plain background (a) | 28 | 207 | 9 | 7 | 36 | 0.2 | 0.24 |

### 3.2.2 PTE ratios of HCl leached fractions

Table 3 displays the leached fraction results for PTE along depth for the four soil cores (concentrations and ratios (leached/total in %)). The studied PTE present different average of leached ratios, increasing as follows: Cr (7 %), Mo (11 %), Ni (25 %), Zn (43 %), Cu (47 %), Mn (62 %) and Cd (68 %). For all the studied PTE, the lowest leached ratios are calculated for core 3. An opposite trend is observed for core 2. Globally, the leached ratios of PTE present no specific variation on the vertical profiles.


**Table 3: Leached concentrations (mg.kg⁻¹) and leached ratios (leached concentration/total concentration in %) of the studied PTE in the 4 soil cores, after HCl 1M extraction. SD: Standard Deviation. N.d.: not determined. "-" means no data.**

| | | Leached concentrations (mg.kg⁻¹) | | | | | | | Leached ratios (%) | | | | | | |
|---|---|---|---|---|---|---|---|---|---|---|---|---|---|---|---|
| Core | Depth (cm) | Cr | Mn | Ni | Cu | Zn | Mo | Cd | Cr | Mn | Ni | Cu | Zn | Mo | Cd |
| 1 | 0-1 | 2 | 60 | 2 | 4 | 21 | 0.05 | 0.12 | 8 | 59 | 28 | 51 | 54 | 11 | 75 |
| | 1-2 | 2 | 67 | 2 | 4 | 23 | 0.04 | 0.12 | 8 | 64 | 28 | 53 | 56 | 9 | 75 |
| | 2-3 | 2 | 68 | 2 | 4 | 23 | 0.04 | 0.13 | 7 | 62 | 27 | 52 | 52 | 8 | 76 |
| | 3-4 | 2 | 68 | 2 | 4 | 25 | 0.04 | 0.14 | 7 | 63 | 28 | 52 | - | 9 | 74 |
| | 4-5 | 2 | 64 | 2 | 4 | 23 | 0.03 | 0.14 | 7 | 62 | 28 | 51 | 52 | 7 | 82 |
| | 5-6 | 2 | 64 | 2 | 4 | 22 | 0.03 | 0.13 | 7 | 61 | 27 | 53 | 54 | 8 | 72 |
| | 7-8 | 2 | 65 | 2 | 4 | 22 | 0.03 | 0.13 | 8 | 61 | 28 | 53 | 53 | 10 | 65 |
| | 9-10 | - | - | - | - | - | - | - | - | - | - | - | - | - | - |
| | 10-11 | 2 | 60 | 2 | 4 | 20 | 0.02 | 0.13 | 6 | 56 | 25 | 52 | 49 | 6 | 72 |
| | Average | 2 | 64 | 2 | 4 | 22 | 0.04 | 0.13 | 7 | 61 | 27 | 52 | 53 | 8 | 74 |
| | SD | 0 | 3 | 0 | 0 | 1 | 0.01 | 0.01 | 1 | 2 | 1 | 1 | 2 | 1 | 5 |
| 2 | 0-1 | 3 | 166 | 3 | 6 | 35 | 0.07 | 0.26 | 8 | 60 | 25 | 53 | 52 | 13 | 67 |
| | 1-2 | 3 | 163 | 3 | 6 | 34 | 0.07 | 0.24 | 9 | 62 | 26 | 54 | 54 | 15 | 71 |
| | 2-3 | 3 | 167 | 3 | 6 | 33 | 0.06 | 0.24 | 11 | 60 | 26 | 54 | 51 | 29 | 73 |
| | 3-4 | 4 | 163 | 3 | 6 | 34 | 0.04 | 0.22 | 11 | 65 | 28 | 53 | 53 | 17 | 67 |
| | 4-5 | 3 | 160 | 3 | 6 | 27 | 0.04 | 0.21 | 10 | 63 | 26 | 54 | 49 | 24 | 68 |
| | 5-6 | 3 | 151 | 3 | 5 | 25 | 0.03 | 0.21 | 9 | 59 | 26 | 52 | 46 | 12 | 62 |
| | 7-8 | 3 | 161 | 3 | 5 | 27 | 0.03 | 0.22 | 10 | 67 | 27 | 55 | 50 | 14 | 69 |





| | | Leached concentrations (mg.kg$^{-1}$) | | | | | | | Leached ratios (%) | | | | | | |
|---|---|---|---|---|---|---|---|---|---|---|---|---|---|---|---|
| Core | Depth (cm) | Cr | Mn | Ni | Cu | Zn | Mo | Cd | Cr | Mn | Ni | Cu | Zn | Mo | Cd |
| | 9-10 | - | - | - | - | - | - | - | - | - | - | - | - | - | - |
| | 10-11 | 3 | 154 | 3 | 5 | 23 | 0.03 | 0.21 | 10 | 64 | 27 | 56 | 49 | 14 | 64 |
| | Average | 3 | 161 | 3 | 6 | 30 | 0.05 | 0.23 | 10 | 63 | 26 | 54 | 50 | 17 | 67 |
| | SD | 0 | 5 | 0 | 0 | 4 | 0.02 | 0.02 | 1 | 3 | 1 | 1 | 2 | 6 | 3 |
| 3 | 0-1 | 4 | 222 | 5 | 4 | 43 | 0.14 | 0.26 | 6 | 58 | 21 | 41 | 48 | 10 | 62 |
| | 1-2 | 4 | 208 | 4 | 4 | 35 | 0.09 | 0.26 | 6 | 62 | 23 | 43 | 48 | 10 | 62 |
| | 2-3 | 3 | 192 | 3 | 3 | 27 | 0.09 | 0.24 | 5 | 58 | 19 | 38 | 40 | 11 | 60 |
| | 3-4 | 3 | 196 | 3 | 3 | 22 | 0.08 | 0.23 | 5 | 61 | 19 | 40 | 39 | 11 | 59 |
| | 4-5 | 3 | 195 | 3 | 3 | 20 | 0.08 | 0.22 | 6 | 61 | 21 | 40 | 38 | 12 | 58 |
| | 5-6 | 3 | 186 | 3 | 3 | 17 | 0.06 | 0.20 | 5 | 60 | 19 | 39 | 34 | 10 | 61 |
| | 7-8 | 3 | 178 | 3 | 3 | 15 | 0.06 | 0.20 | 6 | 60 | 19 | 38 | 32 | 10 | 59 |
| | 9-10 | - | - | - | - | - | - | - | - | - | - | - | - | - | - |
| | 10-11 | 3 | 177 | 3 | 2 | 14 | 0.06 | 0.18 | 6 | 58 | 20 | 35 | 31 | 14 | 60 |
| | Average | 3 | 194 | 3 | 3 | 24 | 0.08 | 0.22 | 6 | 60 | 20 | 39 | 39 | 11 | 60 |
| | SD | 0 | 14 | 1 | 0 | 10 | 0.02 | 0.03 | 0 | 2 | 1 | 2 | 6 | 1 | 1 |
| 4 | 0-1 | 3 | 191 | 4 | 4 | 22 | 0.05 | 0.17 | 7 | 64 | 30 | 43 | 39 | 8 | 71 |
| | 1-2 | 3 | 178 | 4 | 4 | 20 | 0.05 | 0.17 | 7 | 63 | 28 | 42 | 26 | 8 | 74 |
| | 2-3 | 3 | 178 | 3 | 4 | 17 | 0.04 | 0.17 | 7 | 63 | 27 | 41 | 37 | 8 | 71 |
| | 3-4 | 3 | 179 | 3 | 4 | 16 | 0.03 | 0.16 | 7 | 61 | 27 | 43 | 36 | 7 | 70 |
| | 4-5 | 3 | 186 | 3 | 4 | 14 | 0.03 | 0.15 | 6 | 64 | 27 | 42 | 33 | 8 | 65 |
| | 5-6 | 3 | 177 | 3 | 4 | 13 | 0.03 | 0.15 | 7 | 61 | 25 | 40 | 31 | 10 | 71 |
| | 7-8 | 3 | 175 | 3 | 3 | 13 | 0.02 | 0.14 | 7 | 63 | 26 | 41 | 32 | 7 | 67 |
| | 9-10 | - | - | - | - | - | - | - | - | - | - | - | - | - | - |
| | 10-11 | 3 | 178 | 3 | 3 | 13 | 0.01 | 0.15 | 8 | 65 | 26 | 38 | 33 | 4 | 75 |
| | Average | 3 | 180 | 3 | 4 | 16 | 0.03 | 0.16 | 7 | 63 | 27 | 41 | 35 | 8 | 70 |
| | SD | 0 | 5 | 0 | 0 | 3 | 0.01 | 0.01 | 0 | 1 | 2 | 1 | 3 | 2 | 3 |
| | Total average | 3 | 150 | 3 | 4 | 23 | 0.05 | 0.18 | 7 | 62 | 25 | 47 | 44 | 11 | 68 |
| | SD | 1 | 52 | 1 | 1 | 8 | 0.03 | 0.05 | 2 | 2 | 3 | 7 | 9 | 5 | 6 |

## 4 Discussion

### 4.1 Qualitative reactivity of soil cores toward PTE

#### 4.1.1 Reactivity of organic matter in the studied soils

The calculation of hydrogen (HI) and oxygen (OI$_{RE6}$) indexes (Table S4) allows the approximation of the bulk chemistry of the soil organic matter (Espitalié et al., 1977; Vandenbroucke and Largeau, 2007; Saenger et al., 2013). As each biological component (proteins, lignins, lipids, humic and fulvic acids…) is characterized by a particular location within the Van Krevelen diagram (H/C vs O/C ratios) (Balaria et al., 2009; Falsone et al., 2012; Preston and Schmidt, 2006), the position of the studied

samples in the pseudo Van Krevelen diagram (HI : OI$_{RE6}$) indicates their approximate bulk chemistry. Although HI and OI$_{RE6}$ values present slight heterogeneity, the distribution of points on the pseudo Van Krevelen diagram reveals that the organic carbon detected in the four soil cores globally occurred as fulvic acids (Fig. 3) (Saenger et al., 2013). This point may be explained by the presence in the studied soils of calcium carbonate which is considered to stabilize lowly polymerized humic substances such as fulvic acids (Duchaufour, 1970; Duchaufour et al., 2020). This kind of humic substance is particularly



soluble and has an adsorption capacity of metals 2-20 times higher than humic acids, because of a great amount of reactive

functional groups (carboxyl, phenolic, carbonyl…) (Donisa et al., 2003; Borůvka and Drábek, 2004; dos Santos et al., 2020).

However, previous studies showed that pH values ranging from 6 to 8 tend to stabilize metal ions in soils by forming water-

insoluble acid fulvic complexes (Schnitzer and Kerndorff, 1981; Boguta and Sokołowska, 2020). These results suggest a

potential stabilization of PTE inputs by the organic matter of the studied soils and in the current acido-basic conditions (average

pH value : 7.8). Thus, a significant evolution of the pH values of the soils could favour the mobilization of PTE.

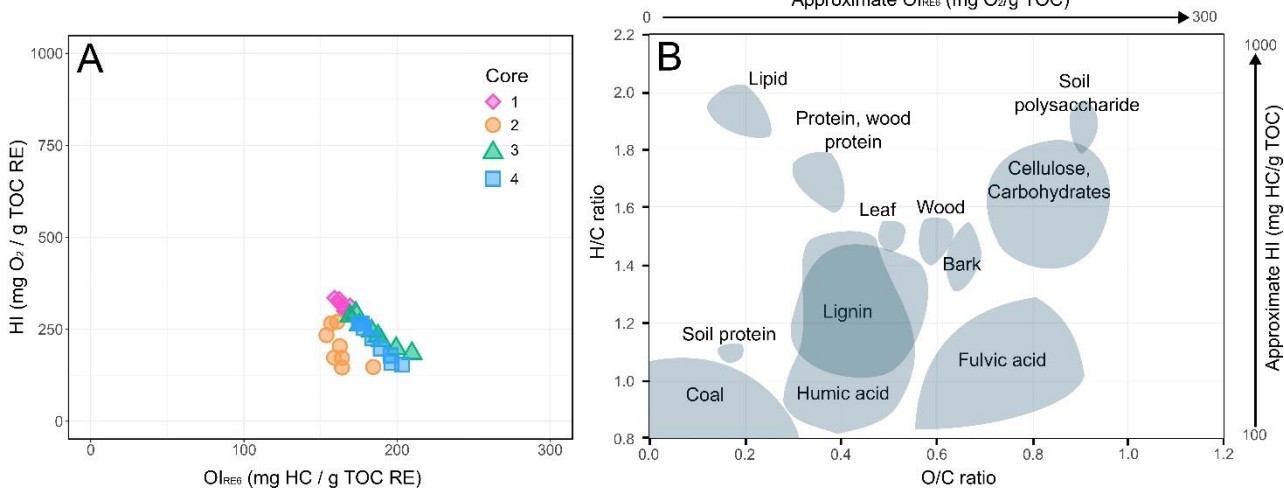

**Figure 3: (A) Position of soil samples in the pseudo Van Krevelen diagram (HI vs. OI$_{RE6}$) according to the different soil cores. (B) location of soil molecules and biological compounds in the Van Krevelen diagram (H/C vs. O/C) and approximate correspondence in the pseudo Van Krevelen (Balaria et al., 2009; Falsone et al., 2012; Preston and Schmidt, 2006).**

**4.1.2 Discrimination of cores by the reactivity of their absorbent complex**

The physicochemical parameters of soil samples revealed significant differences between the soil cores and depths. An

understanding of the nature of the soil cores is required, as these differences may influence the capacity of soil matrices to

associate metallic contaminants and thus their fate in the environment (Fijałkowski et al., 2012; Campillo-Cora et al., 2020;

Yu et al., 2023). For this purpose, a Principal Component Analysis (PCA) was performed on the physicochemical data set.

PCA showed that the two first principal components contributed to 78.4 % of the total variance (Fig. 4). The first principal

component (PC1, 56.2 %) is mainly formed by SiO$_2$, Al$_2$O$_3$, CaO and Fe$_2$O$_3$ contents and discriminates cores by their global

mineralogical composition, as previously described: sandy loam texture with a dominance of SiO$_2$ and quartz, in cores 1 and

2; Silty loam texture with higher Al$_2$O$_3$, Fe$_2$O$_3$, CaO concentrations and clayey fractions, in cores 3 and 4. Besides, illite is

dominant in cores 1 and 2 whereas smectite is the main clay mineral detected in cores 3 and 4. Previous studies highlighted

the importance of smectite in soil absorption capacities (Varadachari et al., 1994; Hanna et al., 2009; Orucoglu et al., 2022) by

their specific surface area and their influence on CEC values (Otunola and Ololade, 2020). The second principal component

(PC2, 22.2 %), composed of TOC, water content and CEC, probably describes the sorption efficiency of the clay-humus soil





complex through organic compounds and the presence of fine-grained materials (Bronick and Lal, 2005; Hernandez-Soriano and Jimenez-Lopez, 2012; Warwick et al., 1998). This second component mostly discriminates subsurface samples from deep

samples on cores 2, 3 and 4, as a higher retention capacity of contaminants characterizes soil surfaces (Chitolina et al., 2020; Fiedler et al., 2007). The two variables allow the discrimination of soil cores and depths according to the potential retention capacity of their absorbent complex (Impellitteri et al., 2002; Bradl, 2004; Lasota et al., 2020). Thus, the subsurface samples of cores 3 and 4 (nearest to the industrial emitters) present the highest sorption capacities compared to the deeper ones. Core 2 (intermediate location between industries and city center) presents the same pattern with globally a lower sorption capacity

than cores 3 and 4, while core 1 (in the city center) presents the lowest sorption capacities with a notable homogeneity with depth.

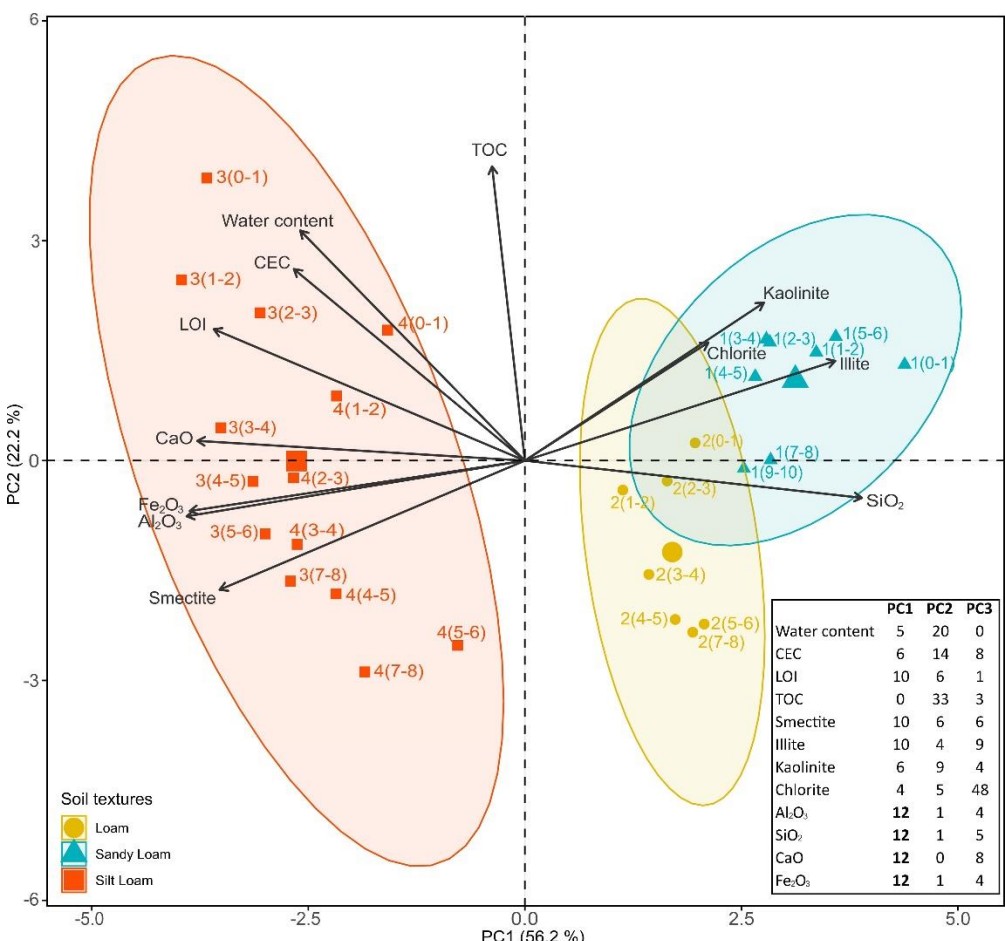

**Figure 4: Principal component analysis (PCA) illustrating the variations of the pedological variables (arrows) for all the soil cores from Gravelines city and textures (colored symbols). Three domains (ellipses) correspond to the different textures of soils and**

**highlight distinct cores, depths and behaviors. Larger symbols indicate the centroid of values for each soil texture. The percentage of contribution of the different pedological variables in the building of the PCA axis is inset (bottom right). The most important contributors are denoted in bold.**





## 4.2 Soil contaminations by industrial or urban PTE inputs

### 4.2.1 Highlighting atmospheric PTE inputs in the soils of Gravelines

Enrichment factors (EF) were calculated for Cr, Mn, Ni, Cu, Zn, Mo and Cd by comparing the soils of Gravelines to those of
the surrounding Wateringues marine plain (Fig. 5) and normalizing to Al contents. Core 1 was not considered because of its
Al concentrations below the detection limit. According to Chen et al. (2007) the cores 2, 3 and 4 can be globally described by
minor PTE enrichments (1<EF<3, Fig. 5) except for Mo (punctual moderately severe enrichments, with 5<EF<10). As
suggested in a previous study (Casetta et al., 2024). The present PTE values and EF support the hypothesis of a diffuse

contamination of the soils of Gravelines. Furthermore, vertical profiles of enrichment factors show that the slight PTE
accumulations in cores 2 (Cr, Mn, Ni, Cu, Zn, Mo, Cd), 3 (Cr, Ni, Zn, Mo) and 4 (Zn, Mo) mostly occur within the two first
cm. As reported in numerous studies (Li and Shuman, 1996; Sterckeman et al., 2000; Williams et al., 1987), these patterns of
superficial accumulation suggest atmospheric and anthropogenic PTE inputs, particularly in cores 2 and 3.

**Figure 5: Vertical profiles of Cr, Mn, Ni, Cu, Zn, Mo and Cd enrichment factors (against Al) in cores 2, 3 and 4. Reference
materials corresponds to the median values obtained on the Wateringues marine plain soils (Sterckeman et al., 2004)**





### 4.2.2 Are the PTE inputs in stations 2 and 3 related to industrial dust?

The previous characterization of dust fallout collected in Gravelines revealed high enrichments in some PTE, compared to the upper continental crust and using Sc as immobile element (Taylor and McLennan, 1995): EF(Cr) = 108, EF(Mn) = 36, EF(Ni)

= 78, EF(Zn) = 60, EF(Mo) = 169, EF(Cd) = 235 (Casetta et al., 2024). As Cu (with EF<20) may originate from multiple sources in such an urban environment (roof, garden and lawn treatment, car brakes; Panagos et al., 2018), it will not be used to trace industrial inputs. Considering these results and other studies carried out on the atmospheric particles emitted in the Dunkerque agglomeration (Alleman et al., 2010; Hleis et al., 2013; Kfoury et al., 2016), a metallurgical dust influence could explain the chemical signatures of the soil cores 2 and 3 from two points of view: (1) the slight but notable accumulation of

Mo, Cr and Ni in the subsurface of soil cores 2 and 3; and (2) the same pattern of accumulation of Mn, Zn and Cd, three other PTE associated with industrial activities. For the core 3, this hypothesis is also supported by the strong correlations calculated between its Mo, Cr and Ni enrichment factors (R>0.93) and its Mn, Cd and Zn enrichment factors (R>0.81). In contrast, these correlations are significantly lower for the core 2. This trend of high correlations between the studied PTE (e.g. R>0.97 for Cr and Ni EF) is globally observed for the all data set (Table 4). Based on this interpretation and on the calculated EF (Fig. 5),

both cores seem to be influenced by Cr, Ni and Mo inputs but core 2 appears more impacted by Mn, Zn and Cd inputs than core 3. Considering Cr as the less mobile PTE in the studied soils (Table 3), the use of some PTE against Cr ratios (Zn/Cr vs 100Cd/Cr and Cr/Mo vs Ni/Mo, Fig. 6) allowed us to visualize and highlight these two different anthropogenic signatures in the considered soil cores, according to depth. Thus, core 3 samples present a chemical signature close to the studied dust data (Fig. 6a). The signature is different for core 2 samples, with the highest considered PTE ratios. Cr/Mo and Ni/Mo ratios (Fig.

6b) underline that the core 2 subsurface samples are marked by the chemical signature highlighted in core 3. Knowing the diversity of atmospheric emissions from metallurgical activities in the studied area (Alleman et al., 2010; Registre des émissions polluantes, 2023; Pollution des sols, SIS et anciens sites industriels, 2023; Hleis et al., 2013), the presence of different industrial chemical signatures in soils is not surprising.

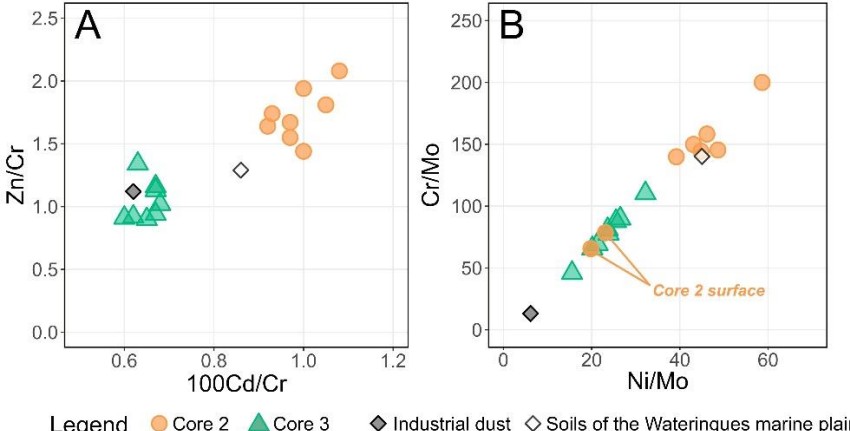

**Figure 6: Elemental ratio diagrams for cores 2 and 3, industrial bulk dust and median Wateringues maritime plain soils. (A) Zn/Cr vs 100Cd/Cr and (B) Cr/Mo vs Ni/Mo.**



No statistically-significant correlation was observed between the calculated EF and the distance from the emission sources, as previously observed with more topsoil samples in Gravelines (Casetta et al., 2024). According to this latter study and regarding the spatial location of the current soil cores, the highest EF measured on core 2 seem to be mostly related to the soil exposure, i.e. the absence of vegetal or urban protecting barriers. Concerning Mn, Cu, Zn and Cd concentrations, urban influences in soils cannot be excluded, considering the potential emission of Mn, Cu and Zn by non-exhaust road traffic (Guo et al., 2012; Lee et al., 2006; Smolders and Degryse, 2002) as well as the correlation between their enrichment factors in soils (R=0.81, Table 4). Likewise, the use of Cu and Cd as pesticide or Mn and Cd occurrence in fertilizers and compost (Baize, 1997; Campos, 2003; He et al., 2005) remains possible.

**Table 4: Spearman's correlation matrix (R values) of pedological parameters, PTE-EF and HCl leached ratios in the studied soil cores (bold values represent significant correlations with R>0.80 and p<0.05). CEC: Cationic Exchange Capacity; TOC: Total Organic Carbon. EF: Enrichment factor.**

| | Water content | CEC | TOC | $Al_2O_3$ | $SiO_2$ | CaO | $Fe_2O_3$ | EF.Cr | EF.Mn | EF.Ni | EF.Cu | EF.Zn | EF.Mo | EF.Cd | % leached Cr | % leached Mn | % leached Ni | % leached Cu | % leached Zn | % leached Mo | % leached Cd |
|---|---|---|---|---|---|---|---|---|---|---|---|---|---|---|---|---|---|---|---|---|---|
| Water content | **1.00** | | | | | | | | | | | | | | | | | | | | |
| CEC | **0.83** | **1.00** | | | | | | | | | | | | | | | | | | | |
| TOC | **0.81** | 0.66 | **1.00** | | | | | | | | | | | | | | | | | | |
| $Al_2O_3$ | 0.07 | 0.23 | -0.30 | **1.00** | | | | | | | | | | | | | | | | | |
| $SiO_2$ | -0.39 | -0.40 | 0.03 | **-0.86** | **1.00** | | | | | | | | | | | | | | | | |
| CaO | 0.17 | 0.25 | -0.24 | **0.92** | **-0.90** | **1.00** | | | | | | | | | | | | | | | |
| $Fe_2O_3$ | 0.07 | 0.23 | -0.35 | **0.94** | **-0.82** | **0.94** | **1.00** | | | | | | | | | | | | | | |
| EF.Cr | 0.08 | -0.10 | 0.20 | -0.47 | 0.09 | -0.14 | -0.41 | **1.00** | | | | | | | | | | | | | |
| EF.Mn | -0.19 | -0.36 | -0.01 | **-0.84** | 0.51 | -0.61 | -0.78 | **0.80** | **1.00** | | | | | | | | | | | | |
| EF.Ni | 0.13 | -0.03 | 0.31 | -0.47 | 0.05 | -0.14 | -0.43 | **0.97** | **0.81** | **1.00** | | | | | | | | | | | |
| EF.Cu | -0.33 | -0.43 | -0.12 | **-0.93** | **0.81** | **-0.89** | **-0.84** | 0.32 | 0.75 | 0.34 | **1.00** | | | | | | | | | | |
| EF.Zn | -0.07 | -0.27 | 0.19 | **-0.82** | 0.50 | -0.64 | -0.80 | 0.70 | **0.90** | 0.73 | **0.81** | **1.00** | | | | | | | | | |
| EF.Mo | 0.77 | 0.59 | 0.79 | 0.03 | -0.51 | 0.28 | 0.03 | 0.58 | 0.20 | 0.60 | -0.13 | 0.29 | **1.00** | | | | | | | | |
| EF.Cd | -0.25 | -0.49 | -0.02 | -0.72 | 0.45 | -0.44 | -0.61 | **0.80** | **0.87** | 0.79 | 0.63 | **0.87** | 0.24 | **1.00** | | | | | | | |
| % leached Cr | -0.67 | -0.67 | -0.39 | -0.45 | 0.61 | -0.54 | -0.40 | -0.07 | 0.44 | -0.07 | 0.79 | 0.47 | -0.60 | 0.40 | **1.00** | | | | | | |
| % leached Mn | -0.16 | -0.05 | -0.05 | -0.02 | 0.19 | -0.17 | -0.00 | -0.46 | -0.19 | -0.49 | 0.28 | -0.08 | -0.52 | -0.20 | 0.47 | **1.00** | | | | | |
| % leached Ni | -0.08 | -0.09 | 0.23 | -0.53 | 0.61 | -0.65 | -0.52 | -0.44 | -0.01 | -0.43 | 0.55 | 0.15 | -0.43 | -0.11 | 0.55 | 0.66 | **1.00** | | | | |
| %leached Cu | -0.37 | -0.50 | -0.04 | -0.72 | 0.69 | -0.67 | -0.64 | 0.28 | 0.66 | 0.26 | **0.86** | 0.74 | -0.15 | 0.63 | 0.76 | 0.31 | 0.49 | **1.00** | | | |
| % leached Zn | -0.12 | -0.31 | 0.28 | -0.77 | 0.70 | -0.73 | -0.76 | 0.53 | 0.71 | 0.52 | 0.65 | 0.78 | 0.12 | **0.81** | 0.49 | 0.05 | 0.41 | 0.76 | **1.00** | | |
| % leached Mo | -0.31 | -0.54 | -0.38 | -0.01 | -0.09 | 0.08 | -0.05 | 0.53 | 0.69 | 0.51 | 0.49 | 0.68 | -0.06 | **0.83** | 0.33 | -0.14 | -0.22 | 0.34 | 0.21 | **1.00** | |
| % leached Cd | -0.08 | -0.02 | 0.32 | -0.57 | 0.62 | -0.75 | -0.64 | -0.41 | 0.05 | -0.34 | 0.49 | 0.15 | -0.38 | -0.22 | 0.38 | 0.37 | 0.69 | 0.36 | 0.36 | -0.44 | **1.00** |

## 4.3 Mobility of PTE in the soils of Gravelines

Assessing the impact of industrial dust on the environment partly stands on the mobility of their dust-borne metals in soils. Thus, the general mobility of PTE in contaminated soils (particularly cores 2 and 3) was measured using 1 M HCl single extraction. Easy to manage, this kind of extraction is often performed on different types of sediment (Hamdoun et al., 2015; Yu et al., 2021) in order to assess the bio-accessible fraction of a specific metal content (Philippe et al., 2008; Roosa et al.,





2016; Snape et al., 2004). According to Billon et al. (2001) and Townsend et al. (2007), HCl is prone to extract the labile fraction (exchangeable, bound to calcium carbonate, part of oxides and to acid volatile sulfides).

Generally, HCl single extraction of bioavailable metal from soils are performed with weaker acid concentrations (0.2 M to 0.5 M) and a shorter contact time (Kubová et al., 2008; Madrid et al., 2007; Pelfrêne et al., 2020). The studied soils developed on

Holocene coastal sediments (deposed during the Gallo-roman Medieval period) are relatively young and thus particularly rich in calcium carbonates (12 to 30 %). This leads us to prefer a higher acid concentration (1 M) and a longer reaction time (24 h), as explains in the protocol used for sediments and formerly presented. This approach allows to avoid buffering effect of the leaching solution by calcium carbonates. It is thus supposed to leach, from the soils, metals potentially mobilized through local changes of pH. Globally, exchangeable metals, metals weakly bound to organic substances (Pelfrêne et al., 2020; Waterlot et

al., 2017), metals precipitated with calcium carbonates or associated with amorphous or poorly crystallized Fe-Mn-oxides or hydroxides (Rao et al., 2010; Yong et al., 1992) are expected to be released by 1 M HCl leaching. To discuss the results of PTE mobility in soils, a table summarizing certain PTE environmental characteristics (uses, environmental sources and pathways, mobility according to environmental conditions), is proposed in Table 5. Considering these data, it is expected in the studied soils a high mobility for Cu, Mn, Mo, Ni, Zn and in a lesser extent Cd, and a low mobility for Cr after HCl 1M

leaching.

**Table 5: Environmental geochemistry of the studied PTE (after Goldberg et al., 1996; Baize, 1997; Reimann and De Caritat, 1998; Crea et al., 2013)**

| Element | Environmental sources and pathways | Mobility in soils | | | |
| --- | --- | --- | --- | --- | --- |
| | | Acidic cond. | alkaline cond. | oxidizing cond. | reducing cond. |
| Cr | Steel works (stainless steel, alloys, chrome plating), electrometallurgy, combustion of natural gas, oil and coal, agriculture (some P-fertilizers) | very low | very low | very low | very low |
| Mn | Steel production, mining and smelting, traffic (antiknock agent in gasoline), agriculture (fertilizers, fungicides), rock weathering, windblown dust | high | very low | very low | high |
| Ni | Steel works (alloys, electroplating), petroleum refining, catalysis, traffic, fuel/coal combustion, agriculture (fertilizers) | high | very low | medium | medium |
| Cu | Cu-mining and smelting, other non-ferrous smelters, electrical industry, plastic industry, steel works (alloys), sewage sludge, agriculture (pesticides, manure), geogenic dust, rock weathering | very high | very low | medium | very low |
| Zn | Zn smelters, galvanizing, alloys, combustion, traffic, waste water, roof, agriculture (pesticides, manure), geogenic dust | high | very low | high | very low |



| Element | Environmental sources and pathways | Mobility in soils | | | |
|---------|-------------------------------------|-------------------|--|--|--|
| | | Acidic cond. | alkaline cond. | oxidizing cond. | reducing cond. |
| Mo | U mining, Mo mining and smelting, alloys, oil refining, oil and coal combustion, sewage sludge, phosphate detergents, agriculture (P fertilizers), geogenic dust, weathering | high | very high | high | very low |
| Cd | Coal combustion, iron and steel mills, electroplating, Pb smelting, incinerators, traffic, sewage sludge, agriculture (fertilizers) | relatively high | medium | variable | very low |

### 4.3.1 Comparison of the different stations

As observed for PTE concentrations, no significant evolution of PTE leached ratios (leached/total content) was observed in the vertical profiles of the four soil cores (according to their SD values) (Table 3). Moreover, PTE leached ratios remain stable even at the subsurface despite the occurrence of organic matter. These results are consistent with the observed lack of correlation between the PTE leached fractions and TOC contents (Table 4). The soil core collected in station 1 (city center) is characterized by the less efficient absorbent complex (Fig. 4) and the lowest total PTE concentrations (Table 2). However, Cu,

Zn and Cd (known as more mobile when coming from mixed anthropogenic sources (Baize, 1997)) were the most extracted PTE in this station (Table 3). Thus, these results reveal a classical urban PTE contamination in the city center (station 1). The soil core collected in station 4 presents higher absorbent capacities (Fig. 4) and higher PTE total concentrations in Cr, Mn, Ni and Mo compared to the soil cores 1 and 2 (Table 2). The correlation between total PTE concentrations may reflect an industrial signature (Cr, Mn, Ni, Mo) on these soils despite the relatively low PTE EF (Fig. 5). Compared to the other cores, the leached

ratios are average for Cr-Mn-Ni and slightly lower for Cu-Zn-Mo (Table 3). This could be related to (1) the soil properties (rich in clay fraction), (2) the relatively lower solubility of PTE from industrial dust or (3) both. The important Cd leached fraction (70 %) is however noticeable. This result is of concern regarding the presence of allotment gardens close to station 4 (Table 3). The highest PTE EF, in particular at surface, are measured in the soil cores 2 and 3 (Fig. 5). The superficial PTE accumulations on these stations (located close to the industrial emitters) were previously related to industrial and atmospheric

inputs (see section 4.2.2). As the present study aims to discuss the ecotoxicological impact of industrial dust deposition on urban soils, the following parts focus on the PTE leached ratios in these cores 2 and 3.

### 4.3.2 Mobility of all the studied PTE in cores 2 and 3

In the soil cores 2 and 3, Mn, Cu, Zn and Cd appear rather mobile compared to Cr, Ni and Mo (Table 3). The mobility of the first group of elements (average leached values both >39 %) in presence of HCl 1M is not surprising considering their natural

mobility in oxidant and/or acidic conditions (Table 5).The high mobility of Mn and Cd (average leached values >60 %) can be additionally explained by (1) their natural association to the exchangeable and calcium carbonates fractions in soils (Table 5, Kubier et al., 2019; Ren et al., 2015), and (2) the concentration of carbonate calcium (easily dissolved by HCl 1M) in these soils (ranging 12 to 28 %). Concerning now the second group of elements, low Cr leached ratios are consistent with the natural





behavior of this PTE in soils (Table 5) and may indicate its association with anthropogenic or natural refractory phases
(Fendorf, 1995). Ni and Mo naturally present strong affinities with calcium carbonates, organic matter and Fe/Mn oxides
(Bielefeldt and Vos, 2014; King et al., 2018; Shi et al., 2012). They are expected to be particularly mobile in oxidant and acidic
conditions (Table 5). Despite the use of HCl 1M treatment, the moderately low Ni and Mo leached ratios (<27 % for Ni and
<17 % for Mo) suggest their association with anthropogenic or natural (iron oxides, silicates) refractory phases (specific
adsorption on organic matter or inclusion in mineralogical phases) (Bibak et al., 1994; Goldberg et al., 1996; Gardner et al.,
2012; Barman et al., 2015).

### 4.3.3 Specific mobility of industrial dust-borne PTE in cores 2 and 3

The four dominant types of industrial dust fallout identified in the city of Gravelines (coal particles, slags, iron ores and
aluminum oxides) are the main bearing phases of several PTE, including Cr, Ni, Mo, Zn, Cd and Mn (Casetta et al., 2024). In
light of current knowledge, it is not possible to relate a PTE to a specific bearing phase. As Cr, Ni and Mo are supposed to be
present in soils within refractory phases, the hypothesis of an industrial nature of these latter in cores 2 and 3 is supported by
(1) the previously described Cr, Ni and Mo enrichment factors and (2) the known low mobility of Cr and Ni found in industrial
coals and slags (Albertsson et al., 2014; Cabrera-Real et al., 2012; Feng et al., 2000; Mombelli et al., 2016; Zhao et al., 2018).
The high mobility of Mn, Zn and Cd in cores 2 and 3 is consistent with previous studies focused on their mobility in industrial
coals and slags (Fernández-Turiel et al., 1994; Querol et al., 1995; Han et al., 2019; Li et al., 2020; Kicińska, 2021). Regarding
the absence of significant correlations between the Mn and Cd EF and their leached ratios, it is however complicated to draw
conclusion about their origin, chemical form and subsequent behavior in the soil cores. Only Zn presents a positive correlation
(R = 0.78), suggesting the higher mobility of this element from anthropogenic inputs. Thus, the study tends to reveal the
stability of industrial dust bearing Cr, Ni and Mo in soils, as these PTE were lowly leached despite the use of a powerful
extraction reagent (HCl 1 M). These results highlight a relative immobility of these harmful elements in the environment
(Smedley and Kinniburgh, 2017; DesMarais and Costa, 2019) and then a low bioavailability.

### 5 Conclusion

The main challenge of this study was to evaluate the vertical distribution and mobility of PTE in the soils of Gravelines, mainly
derived from the deposition of industrial dusts. Although the studied soils globally present minor PTE enrichments, specific
levels of contamination were identified in the soil cores. They were related to industrial dust deposition through (1) the higher
PTE concentrations and EF in the core subsurface (0-3 cm), suggesting anthropogenic and atmospheric inputs of the
contaminants, and (2) the significant associations of metallurgical tracer elements, as Cr-Ni-Mo or Mn-Zn-Cd. The assessment
of the general mobility of industrial PTE in soils reveals the stability of Cr, Ni and Mo, despite the use of a relatively strong
extractant (HCl 1M) and suggests their association to industrial refractory phases. Mn, Zn and Cd have a higher mobility.



Knowing that HCl 1M extraction destabilizes the exchangeable, carbonated, organic and oxide soil fractions, these PTE could
not be only related to metallurgical particles.

The calcareous soils of Gravelines globally present low absorbent capacities, partially counterbalanced by their buffering
capacities. In case of destabilization of industrial dust in soils, these results highlight that the released ions (especially Cr, Ni
and Mo) would be more retained in soils with more efficient absorbent complex and significant carbonate contents (e.g. core
3 vs core 2). Then, the present study shows the importance of studying pedological parameters (texture, mineralogy, TOC,
water content, CEC) to understand their influence on the PTE concentrations and to evaluate their mobility. Evolution of pH
or redox conditions could in fact influence the mobility and then the bioavailability of industrial PTE in the soils of Gravelines.
As these latter are developed at low altitude (0-25 m), their vulnerability in the context of ongoing sea level rise is particularly
significant. They would be more often flooded by seawater or brackish water and thus submitted to variable salinity and/or
redox conditions. These different processes could increase particle weathering (e.g. carbonate, oxides) and consequently induce
higher mobility of some PTE (Hailegnaw et al., 2024; Pellegrini et al., 2024). This hypothesis must be particularly considered,
regarding (1) the potential toxicity of these elements in their mobile form, (2) their accumulation in the soil subsurface, which
interacts with all the environmental compartments and (3) the possible PTE contamination of food produced in the urban
allotment gardens (near the industrial emitters) and consumed by local inhabitants.

**Code/Data availability**

All the data used in this study are presented in the tables and supplementary materials.

**Author contribution**

Marine Casetta: Conceptualization, Data curation, Formal analysis, Investigation, Methodology, Validation, Visualization
Writing—Original Draft preparation, Writing—Review & Editing; Sylvie Philippe: Conceptualization, Funding acquisition,
Investigation, Methodology, Project administration, Resources, Supervision, Validation, Writing—Review & Editing; Lucie
Courcot: Conceptualization, Investigation, Methodology, Supervision, Validation, Writing—Review & Editing; David
Dumoulin: Investigation, Validation, Writing—Review & Editing; Gabriel Billon: Validation, Writing—Review & Editing;
François Baudin: Investigation, Validation, Writing—Review & Editing; Françoise Henry: Investigation, Resources,
Validation, Writing—Review & Editing; Michaël Hermoso: Resources, Supervision, Writing—Review & Editing; Jacinthe
Caillaud: Conceptualization, Funding acquisition, Investigation, Methodology, Project administration, Resources,
Supervision, Validation, Writing—Review & Editing;

**Competing interests**



The authors declare that they have no conflict of interest.

**Acknowledgements**

The leading author thank the Pôle Métropolitain Côte d'Opale (PMCO) and the Region Hauts-de-France for providing a PhD scholarship. The French government and the Region Hauts-de-France are warmly acknowledged for the co-funding of the project (CPER Climibio and CPER MARCO). This study was also supported by the EC2CO project (INSU CNRS) and the SFR Campus de la mer. ICP-AES/MS and XRD analyses were performed on the Chevreul Institute Platform (U-Lille/CNRS) and the Platform CARMIN – Ulille infrastructure, respectively. The authors would like to thank Véronique Alaimo and Marion Delattre for their technical support and assistance in these analyses. Thanks to Laurine Hopf and Louise Legrand who provided helpful technical assistance in the laboratory during their trainees. Authors also thank the agglomeration of Dunkerque, the city of Gravelines, the DREAL, SPPPI and ALOATEC structures.

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
