# Peer review of "A quantitative assessment of the behavior of metallic elements in urban soils exposed to industrial dusts near Dunkerque (Northern France)"

_EGUsphere, 2024_

## Author Response (AR1)

**Author's response to the 1st review**

Dear Anonymous Referee #1,

Thank you very much for your constructive comments on our manuscript. With respect to your suggestions, this one has been revised in order to reorganize the discussion and results sections and clarify each part should we be invited to submit one by the Handling Editor. Please find below the point-by-point responses to your comments for our Preprint egusphere-2024-1875 submission (the following line numbers correspond to the egusphere-2024-1875-marked-up-manuscript-version PDF file). We thank you in advance for your consideration.

Sincerely,

Marine Casetta on behalf of all the co-authors,

**Introduction**

1. L81: "Contamination metals are generally…" - Contamination or contaminant metals?

   Response: The term "contaminant" is more appropriate here but this term has been removed (see next point).

2. L82: "Considering that sources of contamination are multiple, but that some PTE are characteristics of local and industrial dust, the main question concerns their vertical mobility" - Are the PTE coming from local industrial dust or are they both local and from industrial dust? Although I understand the logic and I agree with the above and the following, it's a bit rushed. In what way the local and industrial source of PTE are a reason to study their vertical distribution?

   Response: We have reworked this section to add more details and highlight the link between the sources of PTE and the interest of studying their vertical mobility (L76): "While this previous work revealed the contamination of surface soils by industrial dust, two main questions remain. They relate to the vertical distribution of the metallic contaminants in the soil profiles, and to the capacity of these soils to retain pollutants. These points are crucial, as understanding the behavior of industrial PTE is essential to evaluate their long-term environmental impact, including their potential transfer to groundwater and uptake by biota. The present study is the first to investigate the vertical distribution and possible leaching of PTE (Cr, Mn, Ni, Cu, Zn, Mo, Cd) associated with industrial dust 80 or anthropogenic inputs in selected urban soils (Gravelines)."

3. L92: "were established for better constrain" – to better constrain

   Response: This sentence was corrected L91:" Using these parameters aims to better understand the interactions between PTE and the soil matrix."

**Material and methods**

4. L108: "According to the pedological classification of the French Association for Soils Study, the soils of Gravelines can be ascribed to "Thalassosols", characteristic of a pedogenetic evolution on marine formations (Baize and Girard, 2009; GIS Sol and RMT Sols et Territoires, 2019)." - Perhaps you could add the WRB equivalent of the French classification if it is provided in the AFES 2008 Référentiel pédologique. By definition, Thalassosols develop on marine or fluvio-marine alluvial deposits which is a bit different from 'marine formations', is that what you meant?

Response: We precised "marine or alluvial deposits" instead of "marine formation", and added the international equivalent for our studied soils (i.e. Solonchaks) L108: "According to the pedological classification of the French Association for Soils Study (AFES), the soils of Gravelines can be ascribed to "Thalassosols", characteristic of a pedogenetic evolution on marine or alluvial deposits (Baize and Girard, 2009; GIS Sol and RMT Sols et Territoires, 2019). In the World Reference Base for Soil Resources 110 (WRB) system (FAO, 2014), they can be classified as Solonchaks, reflecting their development in coastal environments."

5. L112: "In the highly populated studied area" – In the/this highly populated study area? I may not have understood

Response: We corrected "the" by "this" highly populated studied area L112. This information aims to explain the presence of a dense channel network.

**Results**

6. From my understanding, conventionally, results are described using past. However, if no other reviewer has pointed this out, ignore my comment.

Response: We would like to keep the present tense to keep the description more alive. There was no comment about this point by the other reviewer.

7. L.270-271: "Maxima CEC values are measured in the subsurface of cores 3 and 4 (7.3 and 7 meq/100 g, respectively) while minimum CEC is measured in core 2 [10-11cm] (3.5 meq/100 g)." – Could you indicate depths for both maximum and minimum CEC value to be more consistent.

Response: Depths were added for maximum CEC values L271: "Maxima CEC values are measured in the subsurface of cores 3 and 4 [0-1cm] (7.3 and 6.8 meq/100 g, respectively) while minimum CEC is measured in core 2 [10-11cm] (3.5 meq/100 g)."

8. Figure 2: It might be better to use two different colour scales between the total mineralogical composition and clay minerals composition since it could be confusing (same color for calcite and illite, quartz and smectite).

Response: The figure 2 (modified in the new version as Figure 3) has been modified with respect to your comment.

9. L318 "…as a global loamy texture…" – Maybe you could be more precise, one of the main features of Thalassosols is that they have a fine grain size (<50μm, as you mentioned: loamy) along the soil profile and continuous presence of carbonates.

Response: We added some details L317 to better justify the classification of soils as thalassossols: "The physicochemical parameters of soils (Table 2) support the AFES pedological classification of the upper soils of Gravelines as "Thalassossols": the samples exhibit significant $CaCO_3$ contents and a predominantly loamy texture along the entire profile. This is consistent with the fine grain size (< 50 μm) typically observed in Thalassossols and in the ploughed soils of the Wateringues marine plain (Sterckeman et al., 2004). In addition, $CaCO_3$ and $CaO$ concentrations are highly correlated (R= 0.97), suggesting a dominance of calcium as carbonates in these soils."

**Discussion**

10. 4.1.1 Reactivity of organic matter in the studied soils (L370) – Why are the results of the organic matter analysis presented in a Discussion? This paragraph is a Results/Discussion paragraph. To be consistent with the chemical and mineralogical description of the soil cores, it would be better to move the description of the Van Krevelen diagram to the Results section, even though it will be quite short, and leave its interpretation to the Discussion section. Perhaps you could spend more time introducing and discussing the differences between core 2 and the other three cores.

    Response: We initially chose to present the Van Krevelen diagram in this section because we considered it as part of the interpretation of HI and OI data. However, taken your comment into account has led us to move the description of the diagram and the associated data to the result section (L285), while the interpretation was maintained in the discussion section (L371). In this latter and as you suggested it, we added some transition with the next part of the discussion to introduce the differences between cores (L387): "While the organic carbon in the studied soils predominantly occurs as fulvic acids (without significant variations between cores and with depth), the pedological properties of the cores (Table 2) reveal notable differences that could influence their reactivity and the behavior of their absorbent complexes."

11. 4.1.2 Discrimination of cores by the reactivity of their absorbent complex (L. 395). Similar comment that for the previous sub-section. It is a mix between Results and Discussion sections.

    Response: We initially considered the description of the PCA as belonging to the discussion, explaining why we inserted it in this section. In accordance with your guidance, we chose to describe the principal components in a new result subsection: "3.1.3 Principal Component Analysis: discrimination of soil cores" (L324), and we developed the differences between cores in the discussion (L395).

12. 4.2.1 Highlighting atmospheric PTE inputs in the soils of Gravelines (L431). This paragraph is a more a result paragraph than a discussion one.

    Response: We chose to present this paragraph in the discussion section because the calculation of enrichment factors involves an interpretation step, resulting of the comparison of our soil PTE concentrations with specific reference values derived from the literature.

13. L 489 - This could be covered first in the methodology and discussed again here, though it is a little surprising to present the entire methodological justification for discussion.

    Response: We initially chose to introduce this section (including Table 5 – modified in the new version as Table 1) in the discussion to enhance readability. With respect to your comment, we have moved this

methodological justification to the Materials & Methods section (L208), along with Table 5 (L228), and reintroduced only a few key notions in the discussion section.

14. Table 5. As this table introduces the potential origin of elements and their mobility in soils, which is one of the main messages of this paper, it should be presented in the introduction or M&M section to justify the studied elements.

    Response: We have moved this table with the previous paragraph, as explained above. In addition, the choice of the studied elements is almost based on the chemical composition of the industrial dust. These elements have been previously highlighted as tracers of industrial activities in the soils of Gravelines (Casetta et al., 2024).

15. 4.3.3 "relative immobility of these harmful elements" - I agree with the low bioavailability, but the introduction and conclusion list other risks such as increased frequency of flooding or use as a garden (children playing with soil, growing root vegetables). What about these risks? Perhaps these risks and potential pathways should be mentioned in the discussion, not just in the introduction and conclusion.

    Response: We developed the last part of the discussion by adding elements about these risks (L556-565) and then slightly modified the conclusion to avoid redundancies (L582).

**Author's response to the 2nd review**

Dear Anonymous Referee #2,

Thank you for your constructive comments on our manuscript. This latter has been revised with respect to your suggestions should we be invited to submit one by the Handling Editor. Please find below the point-by-point responses to your comments for our Preprint egusphere-2024-1875 submission (the following line numbers correspond to the egusphere-2024-1875-marked-up-manuscript-version PDF file). We thank you in advance for your consideration.

Sincerely,

Marine Casetta on behalf of all the co-authors,

1. The abstract lacks a clear structure, e.g., background, methods, results, conclusions.

    Response: As you suggested, the abstract was simplified and sectioned into distinct parts.

2. The introduction should clearly show the knowledge gaps and scientific importance of the study by systematically introducing the published literature.

Response: The introduction has been revised to better emphasize the scientific significance of the study. We have highlighted the limited existing literature on the vertical distribution and mobility of PTE in the study area, and the specific contributions of this work.

3. Line 52: This report indicates that human activities…

Response: This sentence was corrected as you pointed it out.

4. Material & methods: Only four soil cores were collected (at a depth of 11 cm). Is this enough and representative?

Response: The sampling strategy was constrained by several factors: we prioritized the sites previously studied in Casetta et al. (2024a) to benefit from a preliminary understanding of PTE distribution and sources. Additionally, we focused on four sites that were (1) located within the potential deposition area of industrial dust (Figure 1) to ensure their exposure, (2) undisturbed by human activities, (3) publicly accessible, and (4) positioned at discrete distances from emission sources. These criteria were hard to fulfill in this urban context, but necessary for the purpose of the present study. Moreover, while only four soil cores were indeed collected, each core was divided into 11 sections, resulting in over 40 samples, which is sufficient for robust statistical analyses.

5. The discussion lacks depth and logical coherence, which should correspond to the results.

Response: The discussion and results sections have been revised to enhance their structure and logical coherence. Some elements initially included in the discussion, such as the description of the Van Krevelen diagram and the PCA, have been moved to the results section. The justification for using HCl extractions (including Table 5 – modified as Table 1 in the new version) has also been moved to the materials and methods section. Finally, the discussion has been developed on these previous aspects.

6. The conclusion should summarize the key results and mechanisms rather than repeat results or add additional discussion.

Response: Some elements of the conclusion were modified to avoid redundancies with the results and discussion sections, for instance the risks and potential pathways.

---

## Referee Report (RR1)

I would like to thank the authors for their consideration of my comments and for the fact that my comments were taken into account.

The authors have taken my comments into account and modified the manuscript accordingly. Some additions not requested by the reviewers have been made by the authors and improve the quality of the manuscript.

I don't have additional comments and I believe that the manuscript can be accepted for publication as it stands.